# Hematocrit Predicts Poor Prognosis in Patients with Acute Ischemic Stroke or Transient Ischemic Attack

**DOI:** 10.3390/brainsci14050439

**Published:** 2024-04-28

**Authors:** Lingyun Cui, Yefang Feng, Ping Lu, Anxin Wang, Zixiao Li, Yongjun Wang

**Affiliations:** 1Department of Neurology, Beijing Tiantan Hospital, Capital Medical University, Beijing 100070, China; 15836082058@163.com (L.C.); m18810958289@163.com (P.L.); lizixiao2008@hotmail.com (Z.L.); 2The Second People’s Hospital of Huludao, Huludao 125003, China; fengyefang0610@163.com; 3China National Clinical Research Center for Neurological Diseases, Beijing 100070, China; anxin0907@163.com; 4Research Unit of Artificial Intelligence in Cerebrovascular Disease, Chinese Academy of Medical Sciences, 2019RU018, Beijing 100070, China; 5Advanced Innovation Center for Human Brain Protection, Capital Medical University, Beijing 100070, China

**Keywords:** hematocrit, ischemic stroke, transient ischemic attack, outcome

## Abstract

This study aims to investigate the association between HCT (Hematocrit) levels and adverse outcomes in patients with acute ischemic stroke (AIS) or transient ischemic attack (TIA); 14,832 participants from the China National Stroke Registry-III with AIS or TIA were analyzed. Participants were categorized into quartiles based on baseline HCT levels. The primary outcome was poor functional outcomes (modified Rankin Scale ≥ 3) during three months, with secondary outcomes including all-cause death, stroke recurrence, and combined vascular events. Logistic regression or Cox regression models were used to assess the relationship between HCT and clinical outcomes. Compared to the third quartile, patients in the lowest quartile group showed increased risk of poor functional outcome (adjusted OR: 1.35, 95% CI: 1.15–1.58, *p* < 0.001), patients in the lowest quartile had a higher risk of all-cause death (adjusted HR: 1.68, 95% CI: 1.06–2.68, *p* = 0.028), as did those in the highest quartile (adjusted HR: 2.02, 95% CI: 1.26–3.25, *p* = 0.004). Sensitivity analysis shows that the association of HCT with all-cause death weakened, while the association with poor functional outcome was strengthened after excluding patients with recurrent stroke. Our results indicated that HCT level could be used as a short-term predictor for poor functional outcomes and all-cause death in patients with AIS or TIA.

## 1. Introduction

With high rates of disability and fatality, stroke is a complex neurological condition characterized by sudden interruption of blood flow to the brain, leading to tissue ischemia and subsequent neuronal damage [1]. Despite advances in acute management and rehabilitation strategies, Refs. [2,3,4,5,6] ischemic stroke remains a major global health burden, necessitating continued exploration of prognostic indicators to optimize patient care and outcomes [7].

Hematocrit (HCT), the proportion of red blood cells in the blood [8,9], has garnered attention as a potential biomarker for cardiovascular and cerebrovascular diseases [10,11,12,13,14]. Its role in ischemic stroke prognosis has been subject to investigation, with studies suggesting associations between hematocrit levels and outcomes such as mortality, functional disability, and combined vascular events [15,16,17,18,19,20]. The rationale behind this interest lies in hematocrit’s influence on blood viscosity, oxygen-carrying capacity, and hemodynamic stability, all of which may affect cerebral perfusion and tissue oxygenation in the setting of ischemic stroke. However, the findings of these studies are contradictory and the sample sizes are limited.

Therefore, in this study, we aimed to further clarify the relationships between HCT and poor prognosis including poor functional outcome, all-cause death, stroke recurrence, and composite vascular events after acute ischemic stroke (AIS) or transient ischemic attack (TIA) based on the data from the Third China National Stroke Registry cohort (CNSR-III). By elucidating the role of HCT in predicting stroke outcomes, we aspire to aid treatment decisions and ultimately improve patient care and clinical outcomes in the management of ischemic stroke.

## 2. Materials and Methods

### 2.1. Study Design and Subjects

CNSR-III is a national-wide, multi-center clinical registry study of patients suffering from ischemic cerebrovascular diseases. We collected data from 15,166 patients with AIS or TIA who were admitted in the first 7 days of onset in 201 hospitals located in 26 provinces or municipalities in China between August 2015 and March 2018, which have been described previously [21]. After the exclusion of 196 patients without baseline data of HCT and 138 patients without three-month clinical outcome information, 14,832 patients were finally included (Figure 1). The current study was carried out under the guidelines of the Declaration of Helsinki (2013 version) and was approved by the Ethics Committees of Beijing Tiantan Hospital Hospital (approval number: KY2015–001–01). Written informed consent forms were provided by all patients or their legal representatives.

### 2.2. Baseline Data Collection

Baseline information such as sex, age, body mass index (BMI), current smoking, medical history, medication during hospitalization including dehydration treatment, antiplatelet medication, anticoagulant, antihypertensive, and antidiabetic medication, laboratory tests, modified Rankin Scale (mRS) [22] and National Institutes of Health Stroke Scale (NIHSS) [23] was obtained at admission by a trained coordinator following a standard data collection protocol that was developed by the steering committee [21]. Routine laboratory tests including HCT (normal range:39–50%) [24], red cell distribution width (RDW), and leukocytes were obtained by analysis of venous blood samples obtained from patients within the first day of admission. Blood samples were collected, preserved, and processed at each center following standard clinical laboratory policies and procedures.

### 2.3. Outcome Assessment

Participants were interviewed face-to-face at 3 months or by telephone at 1 year by trained coordinators who were blinded to patients’ HCT and other baseline information to assess clinical outcomes after three months. Poor functional outcomes were indicated by an mRS score ranging from 3 to 6. All-cause death was defined as deaths from any causes that could be confirmed by a death certificate issued by the participating hospitals or the local civil registries. Recurrent stroke included three types: ischemic stroke, intracranial hemorrhage, and subarachnoid hemorrhage. Composite vascular events were defined by recurrent stroke, myocardial infarction, all other ischemic vascular events, and all-cause death.

### 2.4. Statistical Analyses

All the selected participants were grouped into quartiles (Q) based on their HCT levels (Q1 < 36.9%, 36.9% ≤ Q2 < 41.1%, 41.1% ≤ Q3 < 44.5%, Q4 ≥ 44.5%) and the third quartile was regarded as the reference group. Baseline characteristics were shown as median plus quartiles or frequency plus percentage, as appropriate. Chi-squared test was conducted for any categorical variables of baseline characteristics. Kruskal–Wallis tests were conducted for any continuous variables. The association of HCT with adverse prognosis was analyzed by logistic regression, with all-cause death, stroke recurrence, and combined vascular events analyzed by the Cox proportional-hazards model. All analyses underwent adjustments from two models. In the first model, we only adjusted for gender and age. In the secondary model, we adjusted all the potential confounders including demographic parameters (age, sex, smoking, BMI, and current smoking), medical history (hypertension, diabetes, hyperlipidemia), mRS score before the onset of index events, NIHSS score at admission, SBP and DBP at admission, medication during hospitalization (antihypertensive, antidiabetic, dehydration treatment), and laboratory tests (Hemoglobin, Red blood cells, Red blood cell distribution width, high-density lipoprotein cholesterol, and platelets) measured at baseline. In addition, we further ruled out patients with recurrent stroke as sensitivity analysis. The interactions of clinically significant characteristics, including sex and age with grouping on the primary outcome’s risk, were also calculated as a subgroup analysis using logistic regression. *p* < 0.05 was considered significant and all *p* values were two-sided. All statistical analyses were carried out with the SAS software (version 9.4, SAS Inc., Cary, NC, USA).

## 3. Results

### 3.1. Baseline Characteristics of Study Participants

Table 1 displays the baseline parameters of 14,832 included participants. Patients with low HCT levels tend to be older, have slightly lower body mass index, a higher proportion of history of hypertension, diabetes mellitus, higher mRS score before the onset of stroke events, lower diastolic blood pressure (DBP), higher NIHSS on admission and more likely to have dehydration treatment during hospitalization compared to those with higher levels of HCT. Compared to the patients with lower HCT levels, those with higher levels of HCT were more likely to be male, smokers, had a higher proportion of history of hyperlipidemia, more likely to have antihypertensive treatment during hospitalization, and have higher baseline hemoglobin, red blood cells, red blood cell distribution width, high-density lipoprotein, and platelet count level.

### 3.2. 3-Month Clinical Outcomes among Patients Grouped by Quartiles of HCT

Table 2 displays incidences of clinical outcomes. In the lowest quartile of HCT, the incidence rates of poor functional outcome, all-cause death, recurrent stroke, and composite endpoint were 16.78%, 2.10%, 6.80% and 7.10%, respectively. In the highest quartile of HCT, the incidence rates of clinical outcomes were 11.65%, 1.22%, 5.37% and 5.55%, respectively. The rate of poor functional outcome, all-cause death, and composite endpoint was highest in the lowest quartile group (*p* = 0.03, *p* < 0.01, *p* < 0.01, respectively), there were no statistically significant differences in recurrent stroke (*p* = 0.05) among groups.

### 3.3. Association between HCT and 3-Month Clinical Outcomes

The correlation between HCT and 3-month adverse clinical outcomes is shown in Table 3. When compared with patients in the third quartile, patients in the first (adjusted OR: 1.35, 95%CI: 1.15–1.58, *p* < 0.01) and second (adjusted OR: 1.86, 95%CI: 1.01–1.39, *p* = 0.035) quartile had significantly increased risk of poor functional outcome, and patients in the first (adjusted HR: 1.68, 95%CI: 1.06–2.68, *p* = 0.028) and fourth (adjusted OR: 2.02, 95%CI: 1.26–3.25, *p* = 0.004) quartile had significantly increased risk of all-cause death. No significant correlation was found between HCT and combined vascular events or recurrent stroke.

We conducted a sensitivity analysis on the correlation between HCT and all-cause mortality as well as poor functional outcomes (Table 4). After further excluding patients with recurrent stroke at three months, the association between HCT and all-cause death weakened, only patients in the highest (adjusted OR: 1.24, 95%CI: 1.03–1.49, *p* = 0.024) quartile of HCT had increased risk. However, patients in the both lowest (adjusted OR: 1.31, 95%CI: 1.10–1.57, *p* = 0.003) and highest (adjusted OR: 1.24, 95%CI: 1.03–1.49, *p* = 0.024) quartile of HCT levels were significantly associated with poor functional outcomes. Subgroup analyses of the outcomes (Table 5) revealed that the association between HCT and adverse clinical outcomes did not differ by age stratification (*p* for interaction > 0.05). However, in the gender stratification, the association between HCT and all-cause death was significant only in males (Q1: adjusted HR: 2.75, 95%CI: 1.50–5.06; Q2: adjusted HR: 1.922, 95%CI: 1.013–3.650; Q4: 2.35, 95%CI:1.27–4.34, *p* for interaction = 0.015).

## 4. Discussion

Our study found that lower levels of HCT were associated with poor functional outcomes and both higher and lower levels of HCT were associated with all-cause death at three months after index event onset. While, when we excluded patients with recurrent stroke, the association with poor functional prognosis strengthened with both lower and higher quartiles of HCT correlated and the association between IL-1Ra and all-cause death attenuated, with higher quartiles of HCT correlated.

Various mechanisms may underlie these associations. In our study, the first quartile of HCT meets the criteria for anemia which causes low blood oxygen-carrying capacity, perhaps further leading to the hypoxic state [17]. The hypoxic state in the presence of low HCT may not only promote the development of the ischemic semi-dark zone towards infarction but also aggravate the malignant progression of the disease by causing a strong inflammatory response in the organism, which may also contribute to the development of poor prognosis. Further, anemia due to low HCT is more closely associated with malnutrition, renal insufficiency, iron deficiency, and inflammation [25,26], which may also contribute to poor prognosis which may explain how it impacts the development of subsequent poor prognosis [27]. In addition, low HCT level leads to prolonged bleeding time [28,29], which may be due to reduced platelet adhesion and decreased thrombin production, and finally increases the risk of hemorrhagic transformation of infarct, possibly contributing to the development of subsequent tissue injury.

A high HCT state implies high blood viscosity, and the ensuing blood stagnation that is likely to occur will cause a decrease in cardiac beat volume and a decrease in cerebral perfusion exacerbating the ischemic–hypoxic response [30,31], while the resulting impaired collateral vasodilatory response will lead to an increase in infarct size [31]. Further, elevated HCT level indirectly increases platelet adhesion through the activation of adenosine diphosphate, leading to platelet dispersion to the subendothelial surface, and accelerating thrombosis [32,33]. In addition, hematocrit is a risk factor for venous thromboembolism [34]. The above mechanisms may explain how they impact the development of subsequent poor prognosis.

However, prior studies also found both high and low HCT levels were correlated with 3-month poor outcomes (mRS ≥ 3) without excluding individuals with recurrent strokes [19]. Unlike our study, this research included a small sample of Europeans who received intravenous thrombolysis treatment, which may be a significant factor contributing to the differing results. Previous studies found that anemia is significantly related to all-causes death in stroke patients [17,35,36,37]. However, these studies did not exclude the population with recurrent stroke. Our sensitivity analysis shows that the association between HCT and all-cause mortality attenuated upon the exclusion of recurrent stroke cases, which implies that the correlation between HCT and all-cause mortality is mainly attributed to stroke recurrence during the follow-up period, rather than having a significant intrinsic relationship. It is important to note that the judicious selection of different treatment regimens can influence the prognosis of patients with ischemic stroke [38,39]. The complex interactions and adverse effects between antiplatelet therapy, anticoagulation therapy, and interventional treatments may also have varying impacts on patient outcomes [40,41]. Furthermore, early identification of high-risk individuals and adherence to guideline-recommended secondary prevention treatments can reduce stroke recurrence and improve prognosis [42].

Our study is the first Chinese study to investigate the correlation between HCT and adverse outcomes after stroke onset. Meanwhile, we had the largest number of the study population. However, we have some limitations: Firstly, as we selected patients from a large-scale nationwide registry, we have not conducted any power calculation to estimate the sample size, and this study was conducted in China and the results need to be replicated elsewhere. Secondly, we only included baseline HCT levels without any follow-up HCT measurement, while those non-baseline HCT levels may also influence stroke prognosis. Thirdly, the HCT levels measured after admission were also influenced by patients’ pre-stroke organic status such as anemia and tumor, which may interfere with our results as well. Finally, there would be bias for hospitalized patients, as some older patients, those who have other serious diseases, and those who may have distinctive levels of HCT, tend to refuse early admission [26].

## 5. Conclusions

Our results indicated that HCT level could be used as a short-term predictor for poor functional outcomes in patients with AIS or TIA. The finding highlighted the role of HCT in the prognosis of stroke, the ease and convenience of HCT testing in clinical practice would make it more widely available in prognosis judgment and rehabilitation assessment of AIS or TIA.

## Figures and Tables

**Figure 1 brainsci-14-00439-f001:**
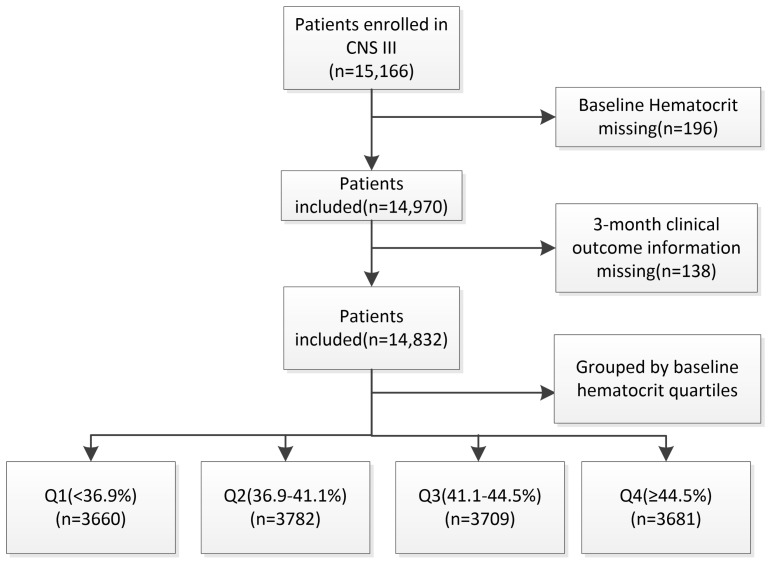
Flow chart of included patients. CNSR III, the Third China National Stroke Registry. Q: quartile.

**Table 1 brainsci-14-00439-t001:** Baseline characteristics of included patients stratified by HCT levels.

Characteristics	Total (n = 14,832)	HCT Level, %				*p*
		Q1 (<36.9) (n = 3660)	Q2 (36.9–41.1) (n = 3782)	Q3 (41.1–44.5) (n = 3709)	Q4 (≥44.5) (n = 3681)	
Demographic parameters						
Age, median, years	62 (54–70)	65 (57–73)	64 (58–72)	62 (54–69)	59 (51–66)	<0.001
Female, n, %	4712 (31.8)	1660 (45.4)	1816 (49.0)	901 (24.5)	335 (8.9)	<0.001
BMI, median, Kg/m^2^	24.5 (22.6–26.6)	24 (22.0–26.1)	24.2 (22.5–26.4)	24.5 (22.9–26.6)	25.0 (23.2–27.0)	<0.001
Current Smoking, n, %	4642 (31.3)	890 (24.3)	802 (21.6)	1245 (33.8)	1705 (45.1)	<0.001
Medical history						
Hypertension, n, %	9277 (62.6)	2336 (63.8)	2370 (63.9)	2222 (60.4)	2349 (62.1)	0.004
Diabetes, n, %	343 (23.2)	912 (24.9)	952 (25.7)	850 (23.1)	722 (19.1)	<0.001
Hyperlipidemia, n, %	1171 (7.9)	249 (6.8)	313 (8.4)	317 (8.6)	292 (7.7)	0.016
mRS score before the onset of index events						<0.001
0–2	14,191 (95.7)	3449 (94.2)	3557 (95.9)	3539 (96.1)	3646 (96.4)	
≥3	641 (4.3)	211 (5.8)	152 (4.10)	142 (3.86)	136 (3.60)	
NIHSS, median (IQR)	3 (1–6)	3 (1–6)	3 (1–6)	3 (1–6)	3 (1–5)	<0.001
Index event						0.146
TIA, n (%)	1004 (6.8)	221 (6.0)	274 (7.4)	252 (6.9)	257 (6.8)	
Ischemic stroke, n (%)	13,828 (93.2)	3439 (94.0)	3435 (92.6)	3429 (93.2)	3525 (93.2)	
SBP at admission, median, mmHg	148 (135–164)	149 (135–165)	148 (135–162.5)	147 (135–161.5)	149 (135–136.7)	0.004
DBP, median, mmHg	86 (79–95)	84 (77–92.5)	85 (77.5–93)	86 (80–95)	90 (80–92)	<0.001
Intracranial stenosis, n (%)	3713 (25.0)	914 (39.8)	956 (39.2)	916 (37.9)	927 (37.7)	0.574
Medication during hospitalization						
Antiplatelet, n (%)	14,303 (96.4)	3526 (96.6)	3575 (97.2)	3547 (97.2)	3655 (97.5)	0.162
Anticoagulants, n (%)	1509 (10.2)	383 (10.5)	379 (10.3)	340 (9.3)	407 (10.9)	0.156
Antihypertensive, n (%)	6841 (46.1)	1651 (45.3)	1684 (45.8)	1663 (45.6)	1843 (49.2)	0.002
Antidiabetic, n (%)	3712 (25.0)	965 (26.5)	978 (26.6)	913 (25.01)	856 (22.8)	<0.001
Dehydration treatment, n (%)	2395 (16.2)	218 (6.0)	188 (5.1)	165 (4.5)	192 (5.1)	0.047
Laboratory test						
HGB, g/L	141 (130–152)	127 (116–144)	131 (127–136)	143 (139–148)	157 (151–164)	<0.001
RBC, 10^×12^/L	4.62 (4.28–4.96)	4.24 (3.9–4.7)	4.32 (4.1–4.5)	4.69 (4.5–4.9)	5.11 (4.9–5.4)	<0.001
RDW, %	42.3 (40.1–44.9)	42.4 (40–45)	42 (39.8–44.5)	42 (40–44.5)	42.9 (40.7–45.2)	<0.001
LDL, mmol/L	2.31 (1.7–3.0)	2.27 (1.7–3)	2.32 (1.7–3.0)	2.32 (1.7–3.0)	2.33 (1.8–3)	0.470
HDL, mmol/L	0.93 (0.78–1.1)	0.94 (0.8–1.1)	0.96 (0.8–1.2)	0.94 (0.8–1.1)	0.90 (0.8–1.1)	<0.001
PLT, 10^×9^/L	212 (175–253)	208 (170–254)	212 (174–254)	212 (177–252)	214 (180–253)	0.002

HCT, hematocrit; BMI, body mass index; mRS, modified Rankin Scale; NIHSS, National Institutes of Health Stroke Scale at admission; IQR, interquartile range; TIA, transient ischemic attack; SBP, systolic blood pressure; DBP, diastolic blood pressure at admission; HGB, Hemoglobin; RBC, Red blood cells; RDW, Red blood cell distribution width; LDL, low-density lipoprotein cholesterol; HDL, high-density lipoprotein cholesterol; PLT, platelet; Q, quartiles.

**Table 2 brainsci-14-00439-t002:** Rates of 3-month outcomes according to quartiles of HCT level.

Outcomes	Total	HCT Level, %				*p*
		Q1 (<36.9) (n = 3660)	Q2 (36.9–41.1) (n = 3782)	Q3 (41.1–44.5) (n = 3709)	Q4 (≥44.5) (n = 3681)	
Poor functional outcome, n (%)	2029 (13.73)	612 (16.78)	544 (14.72)	434 (11.83)	439 (11.65)	<0.001
All-cause death, n (%)	199 (1.34)	77 (2.10)	45 (1.21)	31 (0.84)	46 (1.22)	<0.001
Stroke recurrence, n (%)	927 (6.25)	249 (6.80)	232 (6.26)	243 (6.60)	203 (5.37)	0.052
Combined vascular events, n (%)	959 (6.47)	260 (7.10)	236 (6.36)	253 (6.87)	210 (5.55)	0.033

HCT, hematocrit; Q, quartiles.

**Table 3 brainsci-14-00439-t003:** Association between HCT and stroke recurrence after ischemic stroke or TIA at 3 months.

Outcomes	n (%)	Unadjusted		Age- and Sex-Adjusted		Multivariable-Adjusted ^a^	
		OR/HR ^b^ (95%CI)	*p*	OR/HR ^b^ (95%CI)	*p*	OR/HR ^b^ (95%CI)	*p*
Poor functional outcome							
Q1	612 (16.78)	1.50 (1.32–1.72)	<0.001	1.31 (1.14–1.50)	<0.001	1.35 (1.15–1.58)	<0.001
Q2	544 (14.72)	1.29 (1.12–1.47)	<0.001	1.128 (0.98–1.30)	0.090	1.19 (1.01–1.39)	0.035
Q3	434 (11.83)	Ref.		Ref.		Ref.	
Q4	439 (11.65)	0.98 (0.85–1.13)	0.081	1.14 (0.98–1.31)	0.082	1.16 (0.98–1.37)	0.081
All-cause death							
Q1	77 (2.10)	2.51 (1.6–3.81)	<0.001	1.90 (1.24–2.90)	0.003	1.68 (1.06–2.68)	0.028
Q2	45 (1.21)	1.44 (0.91–2.28)	0.117	1.16 (0.73–1.84)	0.538	1.17 (0.72–1.88)	0.527
Q3	31 (0.84)	Ref.		Ref.		Ref.	
Q4	46 (1.22)	1.45 (0.92–2.28)	0.111	1.85 (1.17–2.93)	0.009	2.02 (1.26–3.25)	0.004
Stroke recurrence							
Q1	2 (6.80)	1.03 (0.87–1.23)	0.727	0.98 (0.82–1.18)	0.856	0.99 (0.82–1.19)	0.905
Q2	232 (6.26)	0.95 (0.79–1.13)	0.537	0.90 (0.75–1.08)	0.264	0.91 (0.75–1.10)	0.321
Q3	243 (6.60)	Ref.		Ref.		Ref.	
Q4	203 (5.37)	0.81 (0.67–0.97)	0.025	0.85 (0.70–1.02)	0.086	0.84 (0.69–1.02)	0.074
Combined vascular events							
Q1	260 (7.10)	1.04 (0.87–1.23)	0.697	0.98 (0.82–1.17)	0.853	0.98 (0.82–1.19)	0.866
Q2	236 (6.36)	0.92 (0.77–1.10)	0.376	0.88 (0.73–1.05)	0.159	0.88 (0.73–1.07)	0.194
Q3	253 (6.87)	Ref.		Ref.		Ref.	
Q4	210 (5.55)	0.80 (0.67–0.96)	0.019	0.85 (0.70–1.02)	0.075	0.84 (0.69–1.02)	0.082

HCT, hematocrit; TIA: transient ischemic stroke; OR: odds ratio; HR: hazard ratio; Ref.: reference. ^a^ In multivariable analysis, adjusted variables included age, sex, smoking, BMI, current smoking, hypertension, diabetes, hyperlipidemia, mRS score before the onset of index events, NIHSS score at admission, SBP and DBP at admission, antihypertensive, antidiabetic, dehydration treatment, and Hemoglobin, Red blood cells, Red blood cell distribution width, high-density lipoprotein cholesterol, platelet. ^b^ All outcomes were evaluated with HR except for poor functional outcome with OR.

**Table 4 brainsci-14-00439-t004:** Association between HCT and 3-month functional outcome in patients without recurrent stroke.

Outcomes	Unadjusted		Age- and Sex-Adjusted		Multivariable-Adjusted ^a^	
	OR/HR (95% CI) ^b^	*p*	OR/HR (95% CI) ^b^	*p*	OR/HR (95% CI) ^b^	*p*
Poor functional outcome						
Q1	1.52 (1.31–1.76)	<0.0001	1.30 (1.11–1.51)	0.001	1.31 (1.10–1.57)	0.003
Q2	1.27 (1.09–1.47)	0.002	1.10 (0.94–1.28)	0.252	1.129 (0.95–1.35)	0.180
Q3	Ref.		Ref.		Ref.	
Q4	0.10 (0.85–1.17)	0.970	1.17 (0.99–1.37)	0.061	1.24 (1.03–1.49)	0.024
All-cause mortality						
Q1	3.29 (1.88–5.76)	<0.0001	2.28 (1.29–4.02)	0.005	1.74 (0.92–3.27)	0.086
Q2	1.55 (0.83–2.90)	0.173	1.17 (0.62–2.20)	0.633	1.09 (0.57–2.09)	0.794
Q3	Ref.		Ref.		Ref.	
Q4	1.32 (0.6–2.52)	0.395	1.78 (0.93–3.41)	0.081	2.11 (1.08–4.09)	0.028

HCT, hematocrit; OR: odds ratio; HR: hazard ratio; Ref.: reference. ^a^ In multivariable analysis, adjusted variables included age, sex, smoking, BMI, current smoking, hypertension, diabetes, hyperlipidemia, mRS score before the onset of index events, NIHSS score at admission, SBP and DBP at admission, antihypertensive, antidiabetic, dehydration treatment, and Hemoglobin, Red blood cells, Red blood cell distribution width, high-density lipoprotein cholesterol, platelet. ^b^ All-cause mortality was evaluated with HR and poor functional outcome was evaluated with OR.

**Table 5 brainsci-14-00439-t005:** Adjusted ^a^ association of HCT with clinical outcomes in each subgroup.

Outcome	Subgroup	HCT				*p* for Interaction
		Q1 (<36.9) OR/HR (95% CI) ^b^	Q2 (36.9–41.1) OR/HR (95% CI) ^b^	Q3 (41.1–44.5)	Q4 (≥44.5) OR/HR (95% CI) ^b^	
Poor functional outcome	Age					
	<65	1.09 (0.87–1.38)	0.96 (0.76–1.22)	Ref.	0.82 (0.64–1.05)	0.301
	≥65	0.86 (0.62–1.20)	0.87 (0.63–1.22)	Ref.	0.93 (0.63–1.38)	
	Sex					
	Male	1.10 (0.87–1.39)	0.92 (0.71–1.18)	Ref.	0.82 (0.65–1.04)	0.431
	Female	0.91 (0.63–1.30)	0.95 (0.68–1.31)	Ref.	0.99 (0.61–1.61)	
All-cause mortality	Age					
	<65	1.88 (0.92–3.83)	1.66 (0.81–3.42)	Ref.	2.10 (1.04–4.22)	0.588
	≥65	1.41 (0.76–2.59)	0.83 (0.44–1.56)	Ref.	1.85 (0.95–3.60)	
	Sex					
	Male	2.75 (1.50–5.06)	1.92 (1.01–3.65)	Ref.	2.35 (1.27–4.34)	0.015
	Female	0.80 (0.38–1.69)	0.55 (0.27–1.11)	Ref.	2.20 (0.99–4.91)	

HCT, hematocrit; OR: odds ratio; HR: hazard ratio; Ref.: reference. ^a^ In multivariable analysis, adjusted variables included age, sex, smoking, BMI, current smoking, hypertension, diabetes, hyperlipidemia, mRS score before the onset of index events, NIHSS score at admission, SBP and DBP at admission, antihypertensive, antidiabetic, dehydration treatment, and Hemoglobin, Red blood cells, Red blood cell distribution width, high-density lipoprotein cholesterol, platelet. ^b^ All-cause mortality was evaluated with HR and poor functional outcome was evaluated with OR.

## Data Availability

The datasets used or analyzed in this study can be obtained from the corresponding author upon reasonable request. The data are not publicly available due to privacy issues.

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
