# Peer review of "Hematocrit Predicts Poor Prognosis in Patients with Acute Ischemic Stroke or Transient Ischemic Attack"

_brainsci, 2024, doi:10.3390/brainsci14050439_

Round 1
Reviewer 1 Report
Comments and Suggestions for Authors
Dear authors, your article is particularly interesting and contains valuable information that could be utilized regarding the prognosis of stroke. Please allow me some questions:
- Does the absence of data on patients' HCT before the stroke pose a threat to your research?
- Did you have any criteria for possibly excluding patients with stroke and other medical conditions that could interfere with or alter the level of HCT?
- The discussion session appears too brief. I suggest expanding it to provide more comprehensive analysis and insights
In the flow chart of included patients, I would suggest checking the numbers. In the first square, you mentioned 15,166 patients àmissing data -196 = included patients (=>15,166-196=14,970?) after that, you missed 138 patients à(14970-138=14832).
In the last square you wrote: Q4(>244.54%) . Is this the last quartile? The number 244.54 refers to the hematocrit?
Paragraph 2.2 Please specify if a standard protocol was used, and if so, it could be helpful to include a sample for reference.
73-79 line: Baseline data was collected in acute phase? Please specify
Paragraph 2.3
Line 81: participants were interviewed face to face or by telephone as a part of a protocol?
Line 83: how did you collect data by phone concerning modified Ranking Scale? Please clarify
Line 97-99: When did you collect information on 'smoking' and 'current smoking'? It's not mentioned above, so clarification would be helpful.
Line 117: The dehydration treatment parameter wasn't mentioned at the beginning of your hypothesis. I suggest incorporating it.
Author Response
Reviewer 1
- Comment: Does the absence of data on patients' HCT before the stroke pose a threat to your research?
Response: Thank you for your valuable feedback. Because HCT serves as a real-time indicator, all patients enrolled in our study experienced onset of symptoms within 7 days, thus We think the lack of pre-stroke data is unlikely to affect our research significantly.
- Comment: Did you have any criteria for possibly excluding patients with stroke and other medical conditions that could interfere with or alter the level of HCT?
Response: Thank you for your valuable review comments. Upon reviewing relevant literature, no established standards for potential interferences or alterations of HCT levels were found. We acknowledge that dehydration treatment prior to onset might influence, yet this data was not recorded in the database. Additionally, anemia could affect HCT levels, but this interrelation with hemoglobin is accounted for in the baseline data.
- Comment: The discussion session appears too brief. I suggest expanding it to provide more comprehensive analysis and insights.
Response: Thank you for your valuable suggestion. We have further supplemented and elaborated on the discussion section. (Line 228-234)
- Comment: In the flow chart of included patients, I would suggest checking the numbers. In the first square, you mentioned 15,166 patients àmissing data -196 = included patients (=>15,166-196=14,970?) after that, you missed 138 patients à(14970-138=14832).
Response: Thank you for your review comments. We appreciate your feedback regarding the errors. We have carefully reviewed your suggestions and corrected the numbers in the flow chart. (Line 69-70)
- Comment: In the last square you wrote: Q4(>244.54%). Is this the last quartile? The number 244.54 refers to the hematocrit?
Response: Thanks for your careful checks. As suggested by the reviewer, we have corrected the “Q4(>244.54%)” into “Q4(≥44.5%)” in the flow chart. (Line 69-70)
- Comment: Paragraph 2.2 Please specify if a standard protocol was used, and if so, it could be helpful to include a sample for reference.
Response: Thank you for your valuable suggestion, a standard protocol was used for data and we have further supplemented the content and the reference. (Line 78-79)
- Comment: 73-79 line: Baseline data was collected in acute phase? Please specify
Response: Thank you for your reminder. For CNSR III, all patients were recruited within 7 days from the onset of symptoms to enrolment, baseline information including prehospital care, pre-stroke modified Rankin Scale (mRS), National Institutes of Health Stroke Scale (NIHSS) score, Age, Blood pressure, Clinical features, Duration of symptoms and presence of Diabetes (ABCD2) score was collected through a direct interview by trained research coordinators at admission. Etiology classification of ischaemic stroke was performed according to the TOAST (Trial of Org 10172 in Acute Stroke Treatment) criteria. Other data were extracted from medical records that include patient demographics, medical history, family history, previous medication, physical examination, primary diagnosis, laboratory tests and risk factor assessment. At discharge, the research coordinators extracted the auxiliary examination and recorded standard aetiological evaluation results, medication, vascular related operation and surgical procedures, final diagnosis, NIHSS and mRS score, economic burden, cerebrovascular events during hospitalisation. In short, all baseline data collection is completed prior to patient discharge. We anticipate that baseline data for the majority of patients would be collected during the acute phase, which we think may not affect our result analysis.
- Comment: Paragraph 2.3
Line 81: participants were interviewed face to face or by telephone as a part of a protocol?
Response: Thank you for your review comments. Participants in our study were interviewed face-to-face at 3 months or by telephone at 1 year and we have made the necessary corrections. (line 85)
- Comment: Line 83: how did you collect data by phone concerning modified Ranking Scale? Please clarify
Response: Thank you for your review, the rained coordinators through telephone follow-up with patients or their caregivers, if the patient has no symptoms at all, an mRS score of 0 is assigned. If the patient, despite having symptoms, shows no significant disability and can carry out all usual activities, an mRS score of 1 is assigned. If the patient has slight disability, meaning unable to carry out all previous activities but can manage personal affairs without assistance, an mRS score of 2 is assigned. If the patient has moderate disability, requiring some help but able to walk without assistance, an mRS score of 3 is assigned. If the patient has moderately severe disability, meaning unable to walk without assistance from others and unable to attend to bodily needs without assistance, an mRS score of 4 is assigned. If the patient is severely disabled, meaning bedridden, incontinent, and requiring continuous nursing care, an mRS score of 5 is assigned. If the patient is deceased, an mRS score of 6 is assigned.
- Comment: Line 97-99: When did you collect information on 'smoking' and 'current smoking'? It's not mentioned above, so clarification would be helpful.
Response: Thank you for your valuable advice, information on 'smoking' and 'current smoking' were collected at baseline. We have updated and added relevant content as per your suggestion. (Line 74)
- Comment: Line 117: The dehydration treatment parameter wasn't mentioned at the beginning of your hypothesis. I suggest incorporating it.
Response: Thank you for your valuable review comments. We consider that the dehydration treatment is one of the adjustment factors for outcomes, incorporating it may affect the results, while,the dehydration treatment parameter, antiplatelet medication, anticoagulant, antihypertensive, and antidiabetic medication were all included in“during hospitalization” in “2.2. Baseline Data Collection”, we have updated and added relevant content. (Line 75-76) We sincerely wish to ask for your opinion. Would this be acceptable?

Reviewer 2 Report
Comments and Suggestions for Authors
This paper reports on the results of a national registry-based study of the effects of haematocrit (HCT) on poor prognosis after acute ischaemic stroke (AIS) or transient ischaemic attack (TIA). This large study showed that lowest quartile HCT had increased risk of poor functional outcome, while lowest and highest quartile had increased risk of and all-cause mortality, at 3 months. The authors felt that HCT could be used as a short-term predictor for outcomes after AIS or TIA.
There are some issues the authors may wish to address:
1. Line 17 – to spell out HCT in full at first use
2. Fig – ‘baseline’ is mis-spelt twice, 41.1 is overlapping, error in >244.54?
3. Line 76 – please provide the normal lab range for HCT
4. Line 81 - were the coordinators blinded to HCT and baseline status?
5. Line 91 – error in 44.54%<Q4?
6. Line 121 – a comma is missing after ‘width’
7. Line 227 – ‘disadvantages’ is better replaced with ‘limitations’. Another limitation may be that this study was done in China, and the results need to be replicated elsewhere
Comments on the Quality of English Languageminor issues
Author Response
Reviewer
- Comment: Line 17 – to spell out HCT in full at first use.
Response: Thank you for your review comments. We appreciate your feedback regarding the spelling errors. We have carefully reviewed your suggestions and made the necessary corrections. (Line 17)
- Comment: Fig – ‘baseline’ is mis-spelt twice, 41.1 is overlapping, error in >244.54?
Response: Thank you for your reminder. We sincerely thank the reviewer for careful reading. As suggested by the reviewer, we have corrected the wrong expressions in Figure 1. (Line 69-70)
- Comment: Line 76 – please provide the normal lab range for HCT
Response: Thanks for your careful checks. As suggested, we have provided the normal lab range(39-50%) for HCT. (Line 80)
- Comment: Line 81 - were the coordinators blinded to HCT and baseline status?
Response: Thank you for your review comments. All the coordinators in our study were blinded to HCT and baseline status. We appreciate your feedback regarding the spelling errors. We have carefully reviewed your suggestions and made the necessary additions. (Line 86)
- Comment: Line 91 – error in 44.54%<Q4?
Response: Thank you for your review comments. We appreciate your feedback regarding the spelling errors. We have carefully reviewed your suggestions and corrected the “44.54%<Q4” into “Q4≥44.5%”. (Line 96)
- Comment: Line 121 – a comma is missing after ‘width’
Response: Thanks for your careful checks. We are sorry for our carelessness. Based on your comments, we have added the necessary comma after ‘width’. (Line 126)
- Comment: Line 227 – ‘disadvantages’ is better replaced with ‘limitations’. Another limitation may be that this study was done in China, and the results need to be replicated elsewhere
Response: Thank you for your valuable feedback. We have replaced ‘disadvantages’ with ‘limitations’ and added the limitation that this study was done in China, and the results need to be replicated elsewhere. (Line 237-239)

Reviewer 3 Report
Comments and Suggestions for Authors
The article, which is an observational study from the China National Stroke Registry-III, evaluates the relationship between hematocrit (HCT) levels and outcomes in patients with acute ischemic stroke (AIS) or transient ischemic attack (TIA). Analyzing a substantial cohort of 14,832 patients, it intriguingly suggests that both low and high HCT levels might be markers of worse prognosis.
The methodology utilized, particularly the categorization of patients into quartiles and the employment of logistic and Cox regression models, is commendable for its effort to provide a structured analysis. Nonetheless, there are several key areas where the study could be refined to enhance its clinical impact and relevance.
One critical aspect that remains unaddressed is the mechanistic explanation behind the observed phenomena. The association of poor functional outcomes with low HCT levels, despite presumably better hemorheologic properties, is counterintuitive and warrants a deeper exploration. Furthermore, the increased mortality in the lowest HCT quartile raises significant clinical concerns, suggesting that an optimal HCT range might be more narrowly defined than previously considered. It is important to note that treatment options inherently influence prognosis after stroke and interactions with antiplatelet anticoagulant and interventional modalities may be explanatory to the findings. However, the review could benefit from a discussion on established pharmaceutical strategies, particularly the use of antiplatelets and anticoagulants in stroke management. A more detailed examination of these conventional treatments, including their benefits and the associated risks of bleeding, would offer a more complete perspective of the field (for recent reviews, see PMID: 34162232).
The sensitivity analysis, which excludes patients with recurrent strokes, seems crucial yet is not fully elaborated upon, particularly regarding its implications for treatment strategies. This could potentially limit the utility of the findings, as understanding the influence of recurrent events is vital for comprehensive stroke management.
A more granular analysis approach could indeed provide greater insights. Segmenting HCT levels into more refined categories such as low, normal, and high, or analyzing HCT as a continuous parameter, might allow for the identification of more precise thresholds that correlate with patient outcomes. Such an approach could lead to more targeted interventions and improve patient care.
Lastly, attention to detail in the presentation of results, such as correcting typographical errors noted in figures and text (e.g., in Figure 1 Q4>244.54%and the term "baslinc"), is essential for maintaining the study's credibility and readability.
In conclusion, while the study makes significant contributions to understanding the prognostic value of HCT in stroke patients, enhancing the analytical framework and providing a mechanistic rationale would make the findings more robust and clinically applicable.
Author Response
Reviewer
- Comment: The association of poor functional outcomes with low HCT levels, despite presumably better hemorheologic properties, is counterintuitive and warrants a deeper exploration.
Response: Thank you for your review comments. In the discussion section of the article, there is a significant amount of text explaining the correlation between low HCT and poor prognosis(Line 196-208). I am so sorry that after multiple readings of the relevant literature, I have not found any additional evidence to support this conclusion. I would greatly appreciate your further guidance if available.
- Comment: It is important to note that treatment options inherently influence prognosis after stroke and interactions with antiplatelet anticoagulant and interventional modalities may be explanatory to the findings. However, the review could benefit from a discussion on established pharmaceutical strategies, particularly the use of antiplatelets and anticoagulants in stroke management. A more detailed examination of these conventional treatments, including their benefits and the associated risks of bleeding, would offer a more complete perspective of the field (for recent reviews, see PMID: 34162232 and PMID: 35976963, and for bleeding risks, see PMID: 36568540 and 32418529).
Response: Thank you for your valuable comments, Your suggestion offers a new perspective and We have further explored the depth of this article from various therapeutic perspectives, while also diligently referencing and citing the four articles you suggested, which have been highly beneficial. (Line 228-234)
- Comment: The sensitivity analysis, which excludes patients with recurrent strokes, seems crucial yet is not fully elaborated upon, particularly regarding its implications for treatment strategies. This could potentially limit the utility of the findings, as understanding the influence of recurrent events is vital for comprehensive stroke management.
Response: Thank you for your review comments. Your suggestion offers a significant perspective and we have expanded our analysis of sensitivity results and discussed the importance of addressing recurrence events for comprehensive stroke management. (Line 232-234)
- Comment: A more granular analysis approach could indeed provide greater insights. Segmenting HCT levels into more refined categories such as low, normal, and high, or analyzing HCT as a continuous parameter, might allow for the identification of more precise thresholds that correlate with patient outcomes. Such an approach could lead to more targeted interventions and improve patient care.
Response: Thank you for your valuable review comments. We initially attempted to group HCT levels as low, normal, and high in our early research phase. However, there was a significant disparity in the sample sizes among the three groups, leading to insufficient statistical power and persuasiveness. We also attempted to categorize HCT as a continuous variable, but found no statistically significant results, which did not support our conclusion, so it was not used. Instead, we divided the data into four groups based on the mean, with the third group approximating the normal range, the first and second groups representing the low-normal range, and the fourth group representing the high range. Considering the results remained persuasive, we opted not to change the grouping. I would greatly appreciate your further guidance if available.
- Comment: Lastly, attention to detail in the presentation of results, such as correcting typographical errors noted in figures and text (e.g., in Figure 1 Q4>244.54%and the term "baslinc"), is essential for maintaining the study's credibility and readability.
Response: Thank you for your review comments. We appreciate your feedback regarding the spelling errors. We have carefully reviewed your suggestions and made the necessary corrections in Figure 1. (Line 69-70)

Round 2
Reviewer 1 Report
Comments and Suggestions for Authors
Dear authors,
I would like to sincerely thank you for your comments, responses, and corrections on the paper. I have no further comments to add at this time. I appreciate the effort and attention you have dedicated to this.